# Fertility Detection of Hatching Eggs Based on a Convolutional Neural Network

**Lei Geng** [1,2] **, Yuzhou Hu** [1,2]**, Zhitao Xiao** [1,2,*] **and Jiangtao Xi** [3]

[1]  School of Electronics and Information Engineering, Tianjin Polytechnic University, Tianjin 300387, China;
    genglei@tjpu.edu.cn (L.G.); huyuzhou1234567@163.com (Y.H.)

[2]  Tianjin Key Laboratory of Optoelectronic Detection Technology and Systems, Tianjin 300387, China

[3]  School of Electrical, Computer and Telecommunications Engineering, University of Wollongong,
    NSW 2522, Wollongong, Australia; jiangtao@uow.edu.au

[*]  Correspondence: xiaozhitao@tjpu.edu.cn

**Abstract:** In order to achieve the goal of detecting the fertility of hatching eggs which are divided into fertile eggs and dead eggs more accurately and effectively, a novel method combining a convolution neural network (CNN) and a heartbeat signal of the hatching eggs is proposed in this paper. Firstly, we collected heartbeat signals of 9-day-later hatching eggs by the method of PhotoPlethysmoGraphy (PPG), which is a non-invasive method to detect the change of blood volume in living tissues by photoelectric means. Secondly, a sequential convolutional neural network E-CNN, which was used to analyze heartbeat sequence of hatching eggs, was designed. Thirdly, an end-to-end trainable convolutional neural network SR-CNN, which was used to process heartbeat waveform images of hatching eggs, was designed to improve the classification performance in this paper. Key to improving the classification performance of SR-CNN is the SE-Res module, which combines the channel weighting unit "Squeeze-and-Excitation" (SE) block and the residual structure. The experimental results show that two models trained on our dataset, with E-CNN and SR-CNN, are able to achieve the fertility detection of the hatching eggs with superior identification accuarcy, up to 99.50% and 99.62% respectively, on our test set. It is demonstrated that the proposed method is feasible for identifying and classifying the survival of hatching eggs accurately and effectively.

**Keywords:** hatching eggs; fertility detection; PhotoPlethysmoGraphy; CNN; heartbeat signal; channel weighting; residual structure

## 1. Introduction

Vaccine–virus immunization is the main method for avian flu pandemic preparedness. Currently bird flu vaccines are generally produced by brewing up live flu virus strains in specific pathogen-free eggs. In the process of virus strain embryos culture, the ability to automatically discriminate the survival of vaccinated eggs prior to virus proliferative cultivation, which can protect the normal eggs from bacterial infections and obtain sterile virus strains, is conducive to removing the dead vaccinated eggs early. If the dead embryos fail to be separated from the hatching eggs early and accurately, the same batch of cultured normal hatching eggs will be contaminated, which will result in major safety and health hazards. Therefore, the fertility detection and classification of the hatching eggs are significant for the production of avian influenza vaccine. Currently, the fertility detection of hatching eggs adopts the method of traditional manual candling by judging the blood vessel characteristics of the embryos. Nevertheless, this method can easily be affected by subjective factors and has the disadvantage of low detection accuracy. In addition, workers who work hard for a long period of time are prone to cause misdetection and missed detection.

At present, there are some popular methods proposed to solve the survival identification and classification problems of hatching eggs. The machine vision methods are widely used in the research of early hatching eggs, which mainly includes image enhancement, segmentation and classification. A non-destructive detection system based on machine vision is designed for identifying the fertility of eggs prior to virus cultivation [1]. Firstly, this method uses the SUSAN (Small Univalue Segment Assimilating Nucleus) algorithm to detect and eliminate the speckle noise. Then a multi-layer feature extraction method is used to segment the blood vessel information in the image of the egg embryos. Finally, the fertility judgment of the hatching eggs is determined by calculating the percentage that the blood vessel area occupies in the ROI (Region of Interest) image. A multi-information fusion method is proposed for detecting the fertility of hatching eggs [2], which fuses the image, temperature and transmittance of embryos and takes it as input for the BP (Back Propagation) neural network to train and classify. Experiments show that the multi-information fusion technology can achieve 96.25% recognition accuracy, however, this method has high requirements on equipment and the operation process is rather complicated. A near-infrared hyperspectral imaging method is proposed to detect the fertility of early hatching eggs [3] by extracting two types of spectral transmission characteristics from the original and Gabor filtered images texture information. However, this method cannot realize multi-classification and the pretreatment process is complicated. High frequency ultrasound imaging was used to detect the development of embryonic cardiovascular tissue [4]. The improved B-mode ultrasound and Doppler effect [5] were used to study the cardiovascular development of egg embryos. However, this method could cause physical damage to the embryos. Zhang et al. [6] propose a method which fuses computer vision technique and the impact excitation technique to increase the detecting accuracy and stability of the fertility of hatching eggs during the early hatching period by adopting the LVQ (Learning Vector Quantization) artificial neural network. Weight Fuzzy C-means clustering algorithm [7] is utilized to find the threshold to segment the blood vessels of the hatching eggs. The fertility is detected by counting the number of blood vessels. A double branches convolutional neural network is proposed to realize the fertility detection and classification of 5-days hatching eggs by using a convolution neural network (CNN) to extract the features of the embryos' blood vessels [8]. The experimental results show that this method successfully solves the multi-classification problem in a small-scale dataset of hatching eggs and obtains a detection accuracy up to 99.5%. Visible transmission spectroscopy combined with multivariate analysis was used to develop a nonde-structive detection system for hatching egg fertility [9], which has a rather high classification accuracy of hatching eggs. However, the effectiveness of multivariate analysis remains to be verified on large-scale datasets of hatching eggs with this method.

These methods mentioned above have greatly improved the fertility detection performance of hatching eggs. However, these methods either cause damage to the embryos or require complicated data preprocessing and have poor detection performance. As an important feature that can directly reflect the survival of animals, a heartbeat signal has the advantages of simplicity, reality and objectivity. In recent years, PhotoPlethysmoGraphy (PPG) has been widely used in the measurement of animal heart rate [10–12]. As a non-invasive means to obtain the heartbeat signal of animals, the noise of the heartbeat signal collected by the photoelectric volume pulse wave tracing technique is less than other acquisition methods and can truly reflect the animal's heart function.

With the rapid improvement of hardware computing power and big data, deep learning is becoming more and more popular. As a representative algorithm of deep learning, the convolution neural network is popular among academia and industry due to the rapid development of deep learning and the outstanding performance in substantial visual tasks. In order to improve the performance of CNN, a lot of excellent CNN models have emerged in recent years and have achieved good performance [13–15]. Furthermore, there are a number of effective measures used for optimizing network structure and improving the classification accuracy of the model. Reference [15] proposed a parameter-modified linear unit that generalized the traditional linear unit and accelerated the model fitting speed with an extra computation cost of almost zero. Reference [16] proposed an operation

named Batch normalization to accelerate the convergence of the model and alleviate the gradient dispersion of the deep network. ResNet [17] has largely settled the problem of vanishing gradients emergence which shackles convergence from the beginning with the network depth increasing. The Squeeze-Excitation (SE) module [18] adaptively reconstructs the channel features by weighting the output of each convolutional layer to improve the network performance.

In this paper, we propose a method to solve the problem of fertility detection and classification of hatching eggs by combining CNN and heartbeat signal of hatching eggs. The main contributions of this work are as follows:

- We propose a deep learning method for the fertility detection of hatching eggs. Our method achieves better performance on survival identification of the hatching eggs thanks to the combination of CNN and the heartbeat signal of hatching eggs.
- We consider the heartbeat signal of hatching eggs as an effective feature to distinguish between fertile eggs and dead eggs, and collect it by using the method of PhotoPlethysmoGraphy to avoid introducing a lot of noise.
- We design a sequence convolutional neural network E-CNN which is used to classify heartbeat sequence of hatching eggs.
- We design a 10-layer-deep convolutional neural network SR-CNN for the fertility detection of hatching eggs by recognizing the heartbeat signal waveform of the embryos, which combines the channel weighting unit "Squeeze-and-Excitation" (SE) and the residual structure to improve the performance of the network.

## 2. Methods

In this section, we first introduce the way of acquiring the experimental heartbeat signal of hatching eggs and the data pre-processing method. Then we illustrate in detail the CNN model structure designed in this paper and explain the strategy that we use to improve network performance.

### 2.1. Data Acquisition and Pre-Processing

The excellent performance of deep learning in the field of classification is based on abundant experimental data distributed independently. In this paper, we choose the heartbeat signal of hatching eggs as the feature data to identify the survival of the embryos. For the purpose of avoiding introducing a large amount of environmental noise during data collection, we adopt the method of PhotoPlethysmoGraphy to collect data. PhotoPlethysmoGraphy is based on Lamber-Bee's law which is described as follows:

$$A = \lg(\frac{I_0}{I}) = \lg(\frac{1}{T}) = kcd \qquad (1)$$

where $I_0$ and $I$ represent incident light and transmitted light intensity respectively, $A$ is absorbance, $T$ is transmittance rate, $k$ is a scale factor, $c$ is concentration of matter, and $d$ is absorbent layer thickness.

We collected heartbeat signal data of 9-day-after hatching eggs by adopting the transmissive light method. The structure of the data acquisition device is shown in Figure 1. In the process of data acquisition, the hatching egg was placed between a laser and a sensor which receives light signals transmitted through the egg. Then the transmitted light signals is converted into digital signals by A/D module. Finally the signal is transferred to the microcontroller via SPI (Serial Peripheral Interface). The 8-second-long heartbeat signal of hatching eggs is sampled at 62.5 Hz continuously for every egg and we eventually get a sequence of 500 data points. The collected heartbeat signal of the embryo is depicted in Figure 2. We can see that the difference of waveform between a fertile egg and a dead egg is not so obvious. Although the waveform of a fertile egg presents a cyclical trend, the amplitude changes are not prominent compared with dead egg. The noise introduced into the signal and baseline drift could account for this phenomenon, and consequently the filtering operation is necessary before processing data.

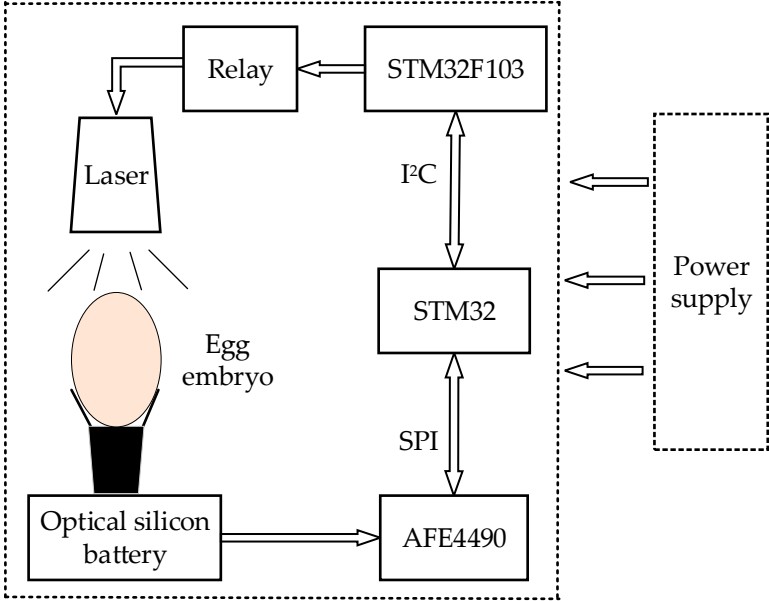

**Figure 1.** Data acquisition device schematic. The system acquisition terminal uses the AFE(Active Front End)4490 chip, which is a fully integrated analog front end designed by Texas Instruments for clinically required arterial oxygen saturation measurements. The control terminal selects the STM (STMicroelectronics)32 microcontroller. The sensor uses a photo-silicon cell with a large photosensitive area, high sensitivity, and wide sensitivity range. The laser source uses a near-infrared source of 808 nm [19].

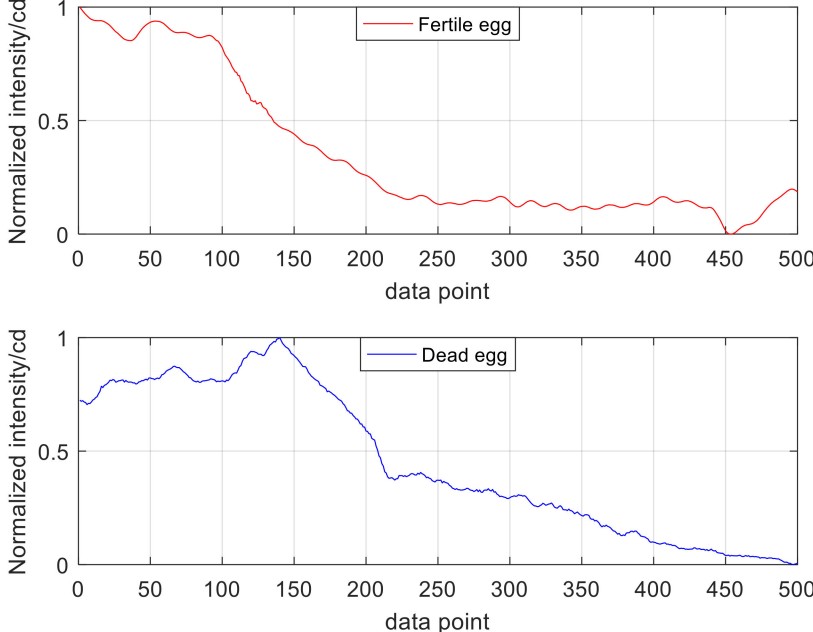

**Figure 2.** The original waveform of collected heartbeat signal of hatching eggs. The amplitude of heartbeat signal data is normalized to within 0–1 in order to reflect the changes of waveform more intuitively.

Given that the heartbeat frequency of 9-day-later hatching eggs ranges from 1 to 4 Hz [20], we design a second order Butterworth high pass filter used for filtering out the noise interference in the original signal and removing the baseline drift. Figure 3 shows the filtered heartbeat signal corresponding to the original heartbeat signal depicted in Figure 2. We take the last 350 stable data of the filtered signal as the sampling points. Apparently, the difference between a fertile egg and a

dead egg on heartbeat signal waveforms becomes more obvious, which is conducive to subsequent classification algorithms.

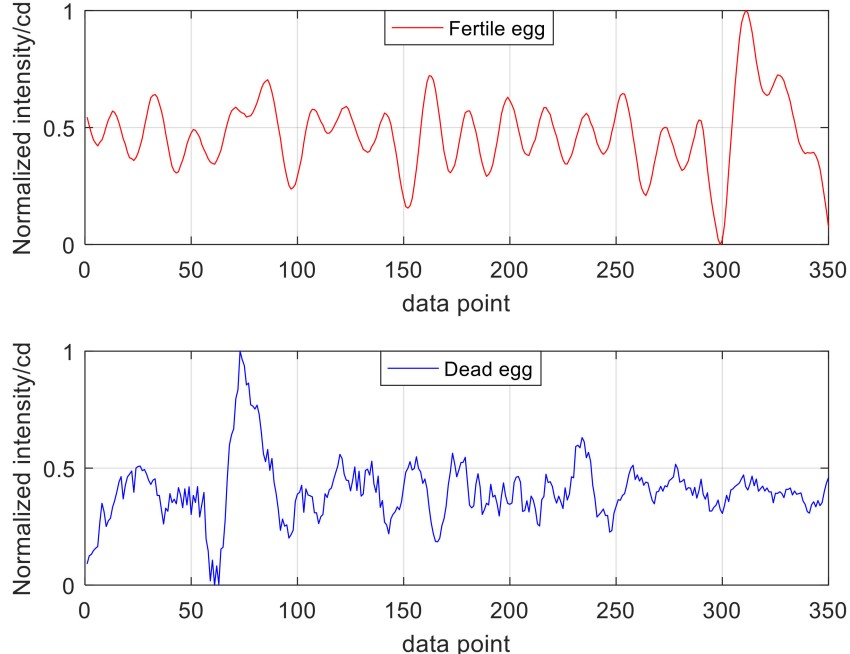

**Figure 3.** The filtered waveform of a collected heartbeat signal of a hatching egg. Only 350 sampling points out of a total 500 are used due to the first 150 data points being unstable after filtering.

## 2.2. Dataset Construction

In deep learning, a suitable dataset is a prerequisite for training a good performance model. The fertility detection of hatching eggs on the 9-days-after injection of avian influenza virus strains is mainly divided into four periods, namely 24 h, 48 h, 64 h, and 88 h after injection. The hatching eggs' heartbeat signals of these four time periods were collected and we got a total of 30,000 data samples. Table 1 lists the sample distributions for different dates in the dataset.

**Table 1.** Experimental dataset sample distribution.

|  | Hatching Eggs Activity Detection Period | | | | Total |
|---|---|---|---|---|---|
|  | 24 h | 48 h | 64 h | 88 h |  |
| Fertile eggs | 3750 | 3750 | 3750 | 3750 | 15,000 |
| Dead eggs | 3750 | 3750 | 3750 | 3750 | 15,000 |
| Total | 7500 | 7500 | 7500 | 7500 | 30,000 |

The datasets used in this paper are divided into two types, one is the hatching eggs' heartbeat sequence dataset, and the other is hatching eggs' heartbeat waveform dataset. The hatching eggs' heartbeat waveform dataset is constructed by converting the collected heartbeat sequence into a grayscale waveform map. The key step in the construction of the waveform dataset is to convert the discrete data points into grayscale waveforms. Figure 4 shows the heartbeat signal waveforms of the dead egg and fertile egg.

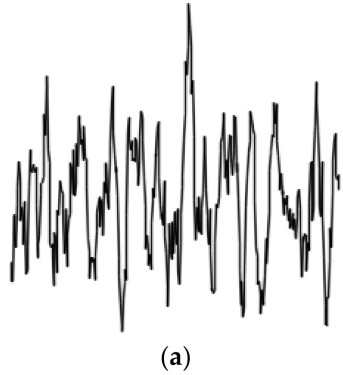
(**a**)

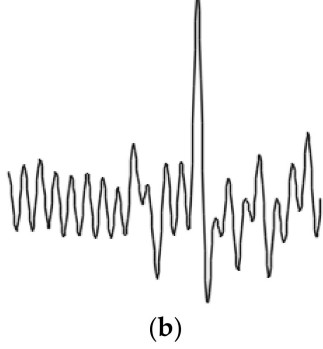
(**b**)

**Figure 4.** The filtered hatching eggs' heartbeat signal grayscale waveform. (**a**) The waveform changes of the dead egg heartbeat signal. (**b**) The waveform changes of the fertile egg heartbeat signal.

It can be seen from Figure 4 that the waveform of the dead egg after conversion is relatively disordered and the change is irregular, while the waveform of the fertile egg is relatively smooth and exhibits periodic beat characteristics. Therefore, the significant difference between fertile eggs and dead eggs on the heartbeat signal grayscale waveform is beneficial for our CNN model to extract the feature.

The sample size of the dataset constructed in this paper is 30,000, and the proportion of positive samples and negative samples is 1:1. The original hatching eggs heartbeat sequence dataset is named DSI (Dataset of Sequential Initials), and the high-pass filtered sequence dataset is named DSH (Dataset of Sequential High-pass filtered). Dataset DWI (Dataset of Waveform Initials) is built by converting the original hatching eggs' heartbeat sequence into a waveform. Dataset DWH (Dataset of Waveform High-pass filtered) is built by converting the high-pass filtered hatching eggs heartbeat sequence into a waveform. N represents the length of heartbeat sequence, and the details of the dataset are shown in Table 2.

**Table 2.** Dataset implementation of DSI, DSH, DWI, DWH.

| Dataset Type | Raw Data (N = 500) | Filtered Data (N = 350) | Total | Positive: Negative | Dataset Partition |
|---|---|---|---|---|---|
| Sequence | DSI | DSH | 30,000 | 1:1 | 8:1:1 |
| Waveform | DWI | DWH | 30,000 | 1:1 | 8:1:1 |

### 2.3. CNN Performance Improvement Strategy

Recently, many works were proposed for the sake of improving the performance of the convolution neural network. At present, people enhance model performance mainly by improving the network structure, such as adjusting the depth and width, and changing the connection mode of the network. As the depth of the network increases, parameter update faces the problem of gradient disappearance. In order to solve this gradient disappearance problem in the deep network, ResNet [17] is proposed. By using the residual structure, ResNet is not only able to solve the network degradation problem but also accelerate network convergence speed by using a "shortcut connection" structure. The structure of the residual block is shown in Figure 5.

Much recent research has sought to better model spatial dependence and incorporate spatial attention. However, SE-Net [18] takes the perspective of feature channels into consideration, which re-balances the relationship between feature channels to improve network performance. SE-Net shows that the "Squeeze-and-Excitation" (SE) block is of great importance when it comes to weighing the relationship between feature channels. The SE block is normally composed of a global pooling layer and two fully-connected layers, it offers two crucial operations called "Squeeze" and "Excitation". The operation of "Squeeze" is used for compressing each two-dimensional feature channels into a real number that represents the global distribution of responses on feature channels. The operation of "Excitation" is a mechanism similar to a recurrent neural network's middle gate for the purpose of

explicitly modeling the correlation between feature channels. The SE block is used to automatically acquire the importance of each feature channel by learning, and then according to this importance to enhance useful features and suppress features that are not useful for the current task. The SE block is depicted in Figure 6.

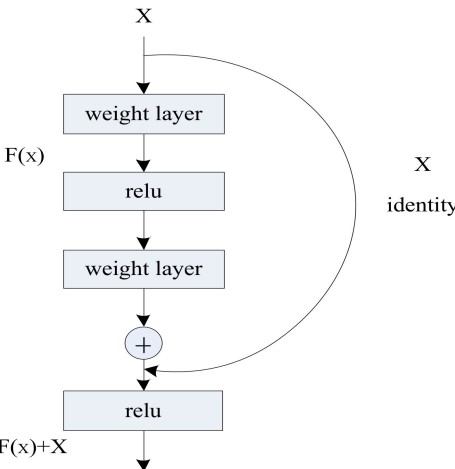

**Figure 5.** The structure of the residual block. It is easier to approximate an identity map by introducing a residual function F(x) = H(x) − X. The "short connection" X introduced makes the gradient back-propagation smoother and reduces the probability of gradient disappearance.

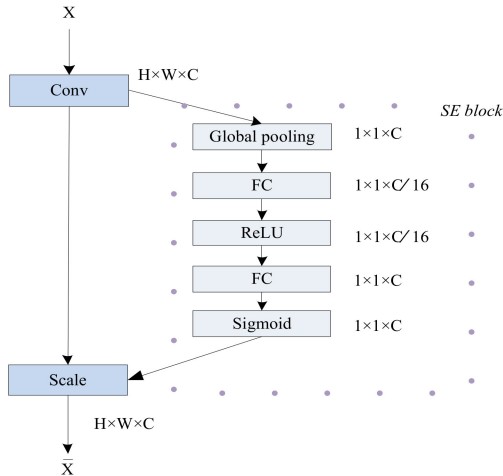

**Figure 6.** The structure of SE block.

### 2.4. Sequence CNN Design

The fertility detection of hatching eggs is a sequence-to-sequence task which takes as its input a heartbeat signal of hatching eggs $X = [x_1, \ldots, x_k]$, and outputs a sequence of labels $r = [r_1, \ldots, r_n]$. For each input sequence $x$, it is corresponding to an element in the tag sequence $r$, the length of input sequence is 350 or 500. Sequence convolutional neural networks are often used to process sequence tasks such as sequence models and natural language processing (NLP). In this paper, we design a sequence convolutional neural network E-CNN used for the classification of 9-days-after hatching eggs' heartbeat sequence.

The E-CNN proposed is mainly composed of five convolution layers, two pooling layers and one fully-connected layer, which are depicted detailedly in Figure 7. The convolution layers all have a filter length of 20 and have 64$k$ filters, where $k$ starts out as 1 and is incremented every 2nd convolution layer. Every convolution layer subsamples its input by a factor of stride, and then max pooling layers subsample their input by using their own stride. The detailed parameter set of E-CNN is shown

in Table 3. We select the rectified linear unit (relu) as our activation function, which is expressed as follows:

$$\text{Relu}(x) = \begin{cases} x \text{ if } x > 0 \\ 0 \text{ if } x \leq 0 \end{cases} \tag{2}$$

What is more, we apply Batch Normalization [16] before each convolutional layer, and we also use Dropout [21] between the convolutional layers and after the non-linearity to prevent network overfitting, the dropout rate is set 0.25 in this paper. The final fully-connected layer and softmax activation produce a distribution over the two output classes for each input sequence.

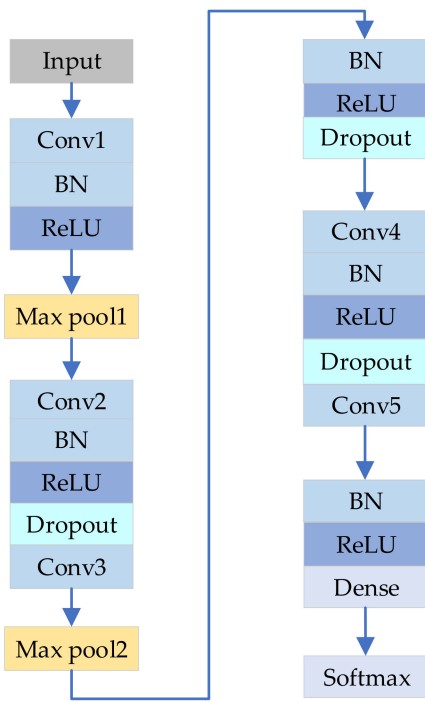

**Figure 7.** The architecture of convolution neural network E-CNN.

**Table 3.** The parameter of model E-CNN.

| Model | Related Parameters | | | | | | | |
|---|---|---|---|---|---|---|---|---|
| | Conv1 | MaxPool1 | Conv2 | Conv3 | MaxPool2 | Conv4 | Conv5 | Dense |
| E-CNN | $1 \times 20$, 64 | $1 \times 5$ | $1 \times 20$, 64 | $1 \times 20$, 128 | $1 \times 5$ | $1 \times 20$, 128 | $1 \times 20$, 256 | 2d |

## 2.5. Two-Dimensional CNN Design

The collected heartbeat signal most contains noises, thus we decided to try to transform the heartbeat signal sequence to a gray waveform image, using the advantage of convolutional neural network in image feature extraction to extract effective features for classification. We propose a convolutional neural network named SR-CNN, which combines SE block and a residual learning unit used for solving the problem that E-CNN has difficulties in extracting the valid features of noisy original heartbeat signals. The channel weighting module SE block measures the dependence between channels by recalibrating the channels of the convolution layer output feature map, making the effective features concerned and the invalid features ignored, so as to make full and effective use of the features contained in waveform image of heartbeat signal. The "shortcut connection" approach used in residual learning unit is not only helpful to solve the performance degradation problem in the deep network but also helps to increase the training speed of the model and improve the effect of network

training. The core for the SR-CNN is a module that we term "SE-Res" (Figure 8a,b), which combines SE block and the residual module for better classification performance improvement.

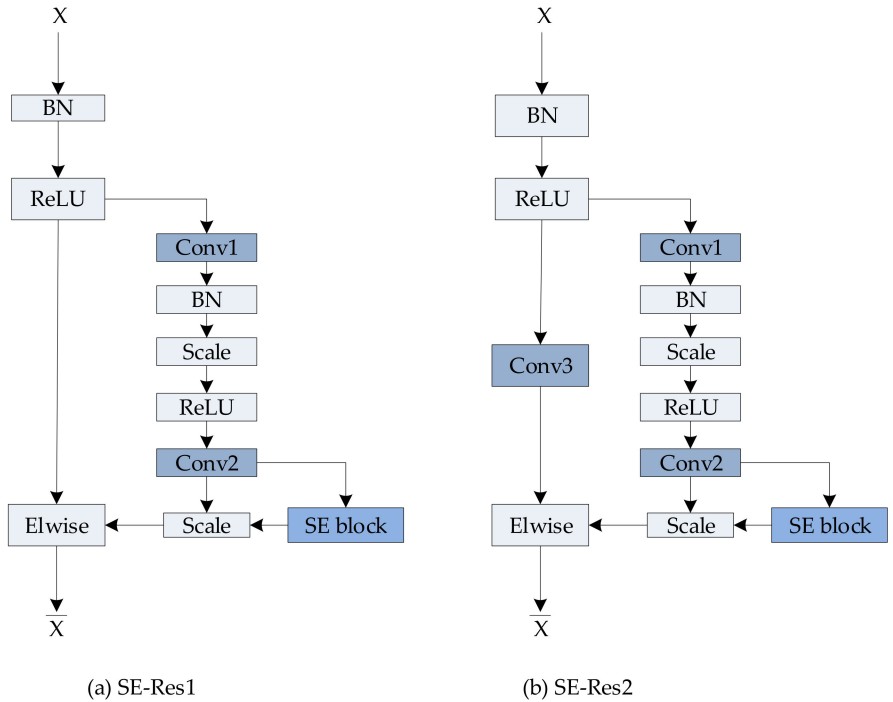

(a) SE-Res1　　　　　　　　　　　　　(b) SE-Res2

**Figure 8.** Two connected schemas of the SE-Res module. (**a**) Represents that the output of two branches have the same channels contrary to (**b**). Here Conv1, Conv2, and Conv3 all represent the convolution layer, where Conv1 has the $3 \times 3$ filter with a stride of 2, and Conv2 has $3 \times 3$ filters with a stride of 1. Besides, the branch of Conv3 with the $1 \times 1$ filter and a stride of 2 is called the "shortcut connection".

SR-CNN mainly consists of one convolution layer, four SE-Res modules, two pooling layers and one fully-connected layer. Before each convolutional layer, we applied Batch Normalization and a rectified linear activation. Figure 9 depicts the network structure, the convolution layers extract low-level heartbeat signal features from the input grayscale waveform, and then the feature maps are normalized by the BN (Batch Normalization) layer to accelerate the learning speed of the network. The Pool1 layer is a max pooling layer that subsamples the feature maps to reduce the size of feature maps. Next, the SE-Res module enhances useful features and suppresses features for the feature maps and its shortcut connections make the optimization of such a network tractable. The Pool2 layer is an average pooling layer; it not only reduces the size of feature maps but reserves more background information for images. The final fully-connected layer and softmax activation produce a distribution over the two output classes (fertile eggs and dead eggs).

It is important to set hyper-parameters and choose an appropriate solution policy for the purpose of reducing the number of parameters and the amount of calculation in the process of model training. Generally speaking, images with higher resolution are helpful for improving network performance; therefore, we set the size of input images as $227 \times 227$ pixels. While training, we followed standard practice and performed data augmentation with random-size cropping to $227 \times 227$ pixels. The convolution kernels that we used with the primary small size of $3 \times 3$ to increase network capacity and reduce parameters. More details and the parameters of different layers are shown in Table 4.

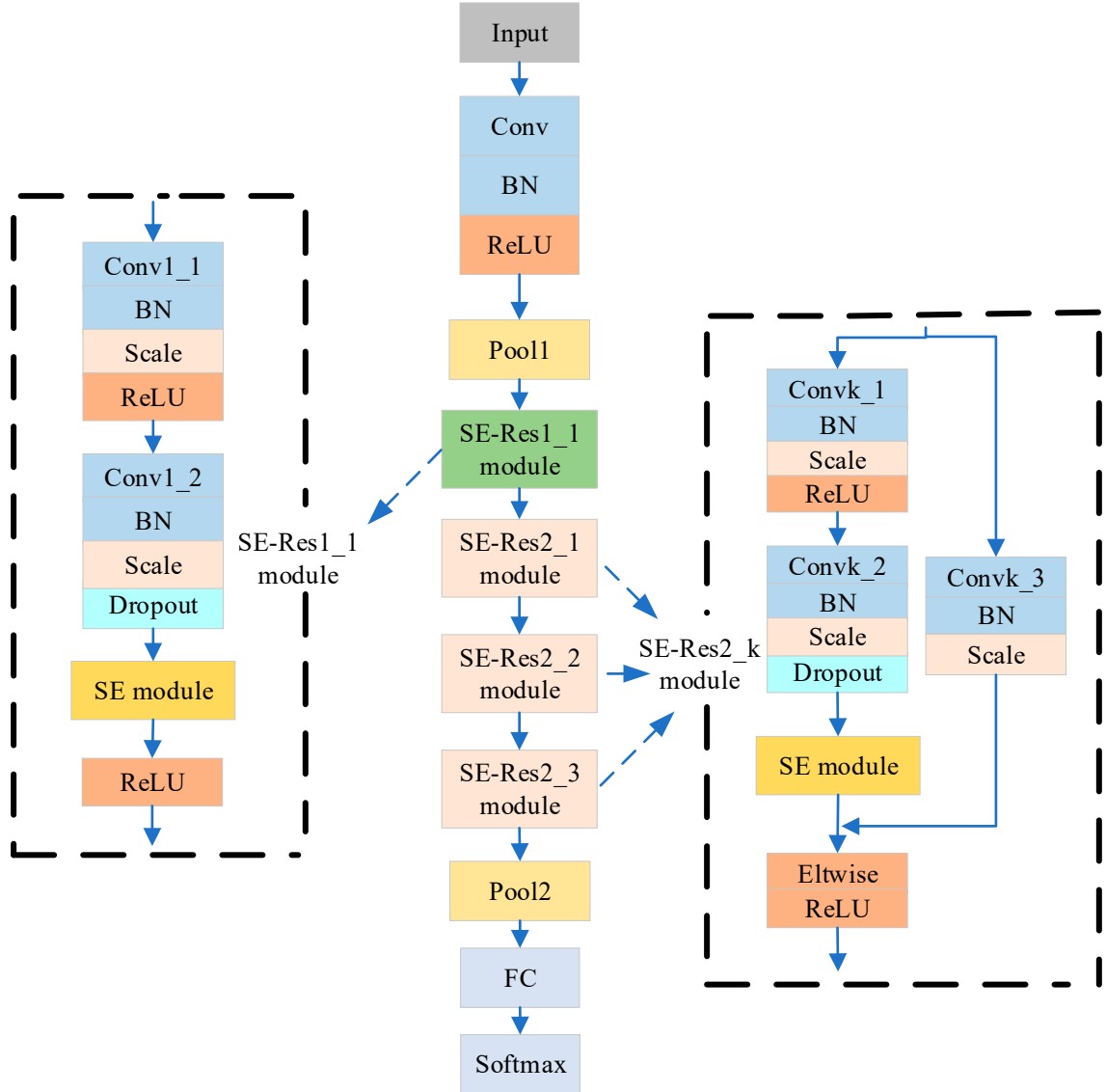

**Figure 9.** The architecture of SR-CNN.

**Table 4.** The parameter of model SR-CNN, C represents the output channel number of the SE-Res module.

| Layer Name | Layer Type | Releated Parameters |
|---|---|---|
| Conv | Convolution | $7 \times 7$, 64, stride 2 |
| Pool1 | Pooling | $3 \times 3$, max pool, stride 2 |
| Conv1_1, Conv1_2 | Convolution | $3 \times 3$, 64, stride 1 |
| Conv2_1, Conv2_2 | Convolution | $3 \times 3$, 128, stride 2, 1 |
| Conv2_3 | Convolution | $1 \times 1$, 128, stride 2 |
| Conv3_1, Conv3_2 | Convolution | $3 \times 3$, 256, stride 2, 1 |
| Conv3_3 | Convolution | $1 \times 1$, 256, stride 2 |
| Conv4_1, Conv4_2 | Convolution | $3 \times 3$, 512, stride 2, 1 |
| Conv4_3 | Convolution | $1 \times 1$, 512, stride 2 |
| Dropout | Dropout | dropout-ratio 0.25 |
| Pool2 | Pooling | $8 \times 8$, average pool, stride 1 |
| FC | Fully connected | 2-d |

## 3. Experiment and Results Analysis

In order to evaluate the effectiveness of the algorithm proposed in this paper, we first trained our network on the dataset built from scratch. Then, we made a detailed analysis of the experimental results.

### 3.1. E-CNN Experiment

In order to verify the validity of the sequence convolutional neural network E-CNN designed before, experiments were conducted on the hatching eggs heartbeat sequence dataset DSI (original signal sequence) and DSH (filtered signal sequence), respectively. We split the dataset into a training set, validation set and test set, the division proportion being 8:1:1. The mini-batch size was set to 64 and the initial learning rate was set to 0.001. We trained E-CNN from scratch and used the Xavier [22] parameter initialization method to initialize weights for the network. In addition, we used the Adam optimizer with the default parameters and reduced the learning rate by a factor of 10 when the validation loss stopped improving. When an epoch was completed in a training set, we tested accuracy once in the validation set. After 150 epochs training, the model come to convergence on dataset DSH, as is depicted in Figure 10. Compared with dataset DSH, E-CNN was unable to reach convergence on dataset DSI, though we have tried many strategies, such as adjusting the number of network layers and parameter settings. We saved the best model as evaluated on the validation set during the optimization process and we obtained a test accuracy rate of 99.50% on the test set.

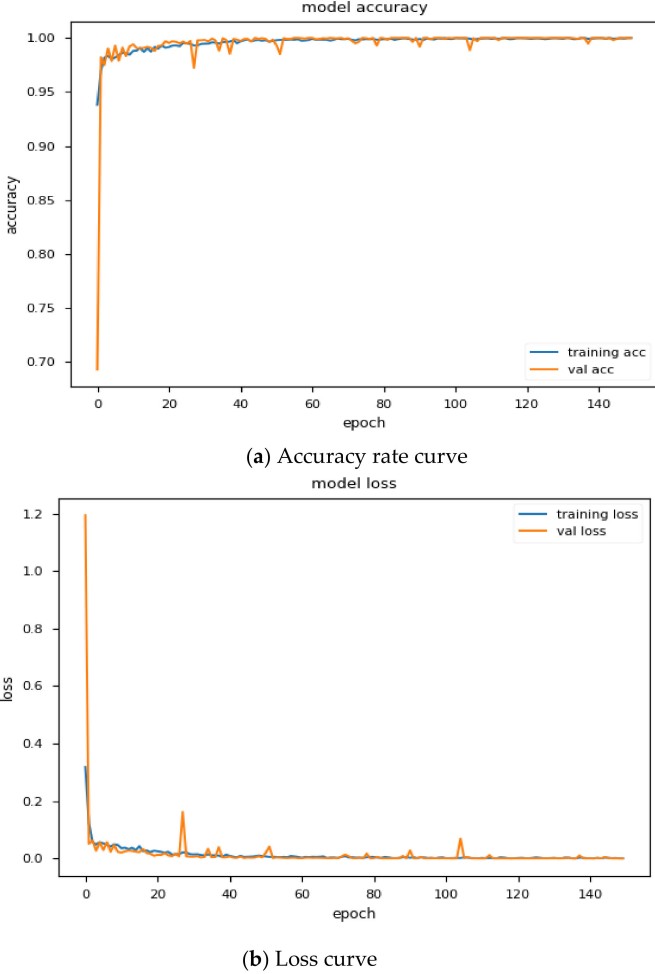

(**a**) Accuracy rate curve

(**b**) Loss curve

**Figure 10.** Accuracy and loss curve of E-CNN on dataset DSH.

### 3.2. SR-CNN Experiment

Knowing from the experiments of E-CNN that it is difficult for sequence convolution to extract features effectively from noisy signal which leads to intractability while training. Thus, we enhanced the dimension of the heartbeat signal by converting sequence into waveform, which is beneficial for SR-CNN to make full use of the waveform while extracting features.

We conducted experiments on dataset DWI (original heartbeat signal waveform) and DWH (filtered heartbeat signal waveform). Before training, it was necessary to set the solution hyperparameter. The mini-batch sizewas set to 64, the momentum coefficient momentum was set to 0.9, the weight attenuation coefficient weight_decay was set to 0.0001 to prevent overfitting. The Stochastic Gradient Descent (SGD) was chosen to update weight. As for learning rate adjustment strategy, we selected the step strategy. The initial learning rate base_lr was set to 0.01, the learning rate change index gamma was set to 0.1, and the learning rate change step was set to 40 epochs (1 epoch represents all pictures of the training set is trained one time). The change formula of the learning rate during training is as follows:

$$lr = base\_lr * gamma^{\left(floor\left(\frac{iter}{stepsize}\right)\right)} \tag{3}$$

Different network architectures are compared to assess the rationality and feasibility of the network we designed. We discuss four different network architectures on our dataset DWH, with the mini-batch size is set to 64 during training. We designed a network architecture named Basic network, which is almost the same as SR-CNN except without using the SE module and "short connection". Another two networks are designed based on Basic network by adding the SE module and the "short connection" respectively. We selected 200 epochs during network training and compared the changes of accuracy and loss; the results are shown in Figure 11.

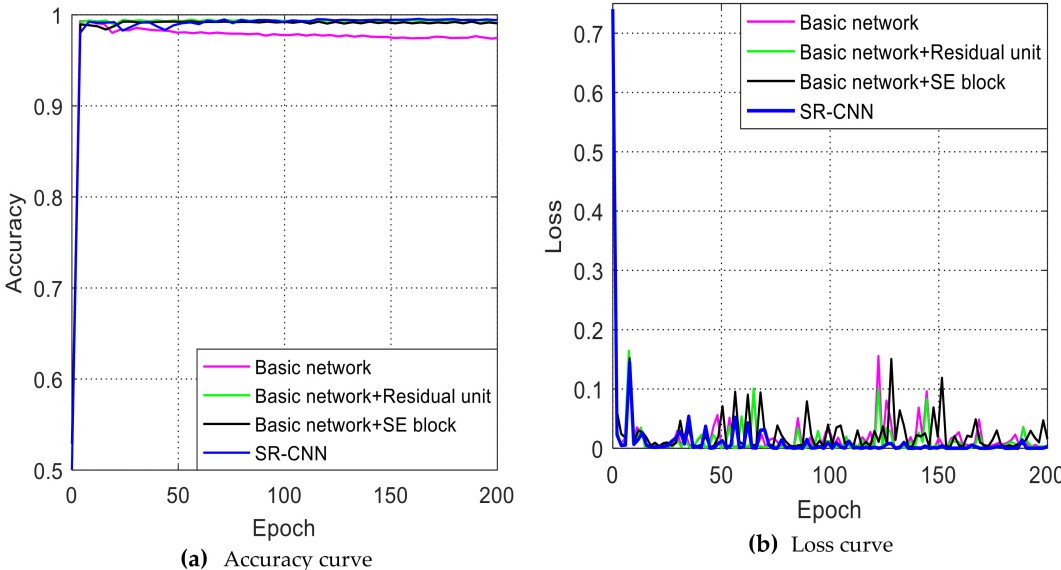

**Figure 11.** Accuracy and loss of different network architectures on dataset DWH.

During the network training, the generated model was assessed on the validation set, and the model as evaluated as the best was saved. In the period of the test, the saved optimal model was selected to test the heartbeat waveform of the test set and the test accuracy of the model was obtained.

### 3.3. Results Analysis

The fertility detection of the hatching eggs is a binary classification problem in this paper. We used not only Accuracy but also use other metrics such as Precision, Recall and F1 score to evaluate

the model's comprehensive performance. We tested different models on the dataset DWH; the test set sample distribution is shown in Table 5.

**Table 5.** Test set sample distribution.

| Sample Type | Detection Period | Number of Sample | Total |
|---|---|---|---|
| Fertile eggs | 24 h | 395 | 1580 |
| | 48 h | 410 | |
| | 64 h | 575 | |
| | 88 h | 200 | |
| Dead eggs | 24 h | 355 | 1420 |
| | 48 h | 421 | |
| | 64 h | 288 | |
| | 88 h | 356 | |

The fertile eggs are defined as positive samples, and the dead eggs are defined as a negative samples. In the test, the number of fertile eggs detected correctly in the samples is defined as TP, and the number of fertile eggs detected falsely is defined as FN. The number of dead eggs detected correctly in the sample is defined as TN, and the number of dead eggs detected as fertile eggs is defined as FP. We the calculated Accuracy, Precision, Recall and F1 scores according to formula (4).

$$\begin{cases} \text{Accuracy} = \frac{TP+TN}{TP+TN+FP+FN} \\ \text{Precision} = \frac{TP}{TP+FP} \\ \text{Recall} = \frac{TP}{TP+FN} \\ \text{F1} = \frac{2*\text{Precision}*\text{Recall}}{\text{Precision}+\text{Recall}} \end{cases} \tag{4}$$

It can be seen from Table 6 that the network model designed in this paper has the best performance on the test set compared with other network models due to the combination of the SE module and the residual block. The model trained by network SR-CNN has the lowest false detection rate and the highest detection accuracy. The Recall and F1 scores are higher than on the other models, which indicates the feasibility and superiority of the SR-CNN network model designed in this paper.

**Table 6.** The performance of different models on the test set.

| Model Name | Sample Distribution | Recognition Result | | False Detection | Accuracy (%) | Precision (%) | Recall (%) | F1 Score (%) |
|---|---|---|---|---|---|---|---|---|
| | | P | N | | | | | |
| Basic network | P (1580) | 1569 | 11 | 22 | 99.28 | 99.30 | 99.30 | 99.30 |
| | N (1420) | 11 | 1409 | | | | | |
| Basic network+ Residual unit | P (1580) | 1570 | 10 | 19 | 99.37 | 99.43 | 99.36 | 99.40 |
| | N (1420) | 9 | 1411 | | | | | |
| Basic network+ SE block | P (1580) | 1573 | 7 | 17 | 99.44 | 99.37 | 99.57 | 99.47 |
| | N (1420) | 10 | 1410 | | | | | |
| SR-CNN | P (1580) | 1575 | 5 | 12 | 99.62 | 99.56 | 99.68 | 99.62 |
| | N (1420) | 7 | 1413 | | | | | |

The SE module models the nonlinear dependence relationship of feature channels by recalibrating the importance of each feature channel, which simplifies the learning process and enhances the network expression ability. The SE module reduces the dimension of the global features to the original 1/R by the stimulus operation, and then reverts to the previous dimension after being activated by the Relu function. In this paper, different values of hyper parameter R are selected to train the model, and then we compare the accuracy of the training model on the test set. The experimental results are shown in Table 7. It can be seen that the SR-CNN has the highest accuracy on the test set when R is 16. Thus, we chose 16 as the multiple of dimension reduction and dimension-enhancement of the fully connected layer.

**Table 7.** Comparison of accuracy with different values of hyper parameter R.

| R | Accuracy (%) |
|----|----|
| 4 | 94.52% |
| 8 | 96.58% |
| 16 | 99.62% |
| 32 | 97.73% |

*3.4. Comparsion with Other Methods*

To illustrate the feasibility of our method, we compared our method with state-of-the-art methods on the classification performance of 9-days-after hatching eggs. Restricted by hardware conditions and the differences in datasets used by various methods, we cannot reproduce these methods on our datasets. Since the dataset used in our method is the heartbeat signal of hatching eggs, which is different from the dataset used in other methods, and we lack the public dataset, thus we only compared the performance on their own datasets. Each approach has its own feature extraction method, so we also make the comparison on the methods of feature extraction. The comparison of experimental results is listed in Table 8.

**Table 8.** Comparison of experimental results.

| The Method | Feature Extraction Method | Classification Method | Date | Accuracy | Classification |
|----|----|----|----|----|----|
| Proposed method | SR-CNN, E-CNN | SR-CNN, E-CNN | 9-day-later | 99.62%, 99.5% | Fertile and Dead |
| Geng, L. et al. (2017) [8] | TB-CNN | TB-CNN | 5-days | 99.5% | Fertile, Dead and Infertile |
| Liu, L. et al. (2013) [3] | Gabor-filter | K-means clustering | 4-days | 84.1%. | Fertile and Non-fertile |
| Shan, B. et al. (2010) [7] | Thresholding by Histogram-based WFCM | Criterions of blood vessels | Middle-stage | 99.33% | Fertile and Non-fertile |
| Md. Hamidul Islam. et al | k-means, LDA, SVM | k-means clustering, LDA, SVM | 4-days | 96%, 100%, 100% | Fertile and Infertile |

From the comparison with state-of-the-art methods, we can find some advatanges of our method. Among these methods, the method of TB-CNN achieved multi-classification and made a great breakthrough on classification accuracy by using the CNN structure which is powerful and effective to extract blood vessel features. However, this method requires complex vascular image labeling before training; besides this, it also needs to further validate the effectiveness of hatching eggs detection in other periods. The near infrared hyperspectral imaging system is based on the connection of the InGaAs camera and line scanning spectrometer, which has a high cost. The method proposed in reference [7] not only is highly robust on image noise, eggshell color, and dirty spots on the eggshell, but also meets the needs of mass production both in detection accuracy rate and executive speed. However, the two methods mentioned above have difficulty in extracting some thin blood-vessel features of hatching eggs' vascular image, which infunences the accuracy of the method. By contrast, the feature we used in this paper, the heartbeat signal of hatching eggs, is simple and we have a lower cost of data acquisition devices. The method of visible transmission spectroscopy combined with multivariate analysis is developed and has the potential to be used for the detection of infertile eggs in commercial hatchery operations with its high detection accuracy. However, the transmission spectroscopy changes with the kinds of hatching eggs, therefore, the effectiveness of multivariate analysis remains to be verified on large-scale datasets of hatching eggs. Our method used the dataset of various periods of hatching eggs, and the heartbeat feature of hatching eggs is stable, which is beneficial for our CNN to extract effective features for classification. Our method is superior to the other methods mentioned above in detection accuracy and algorithm robustness. which benefits from the combination of the CNN model that we designed and the heartbeat signal of hatching eggs.

## 4. Conclusions

In this paper, we proposed a novel method used for the fertility detection of hatching eggs by combining a convolution neural network and the heartbeat signal of hatching eggs. We collected

the heartbeat signals of hatching eggs by the method of PhotoPlethysmoGraphy and preprocessed the collected data to reduce noise interference. We analyzed the collected data from different data dimensions. First, we designed a sequence convolution neural network model E-CNN used for tackling the heartbeat signal sequence. Then, in order to solve the problem that E-CNN is not able to deal with noisy signals, we designed SR-CNN which combines the SE block and the residual learning unit in a convolution neural network to enhance the ability to extract features from the heartbeat signal waveform. Experimental results show that the two network architectures we designed are effective in hatching eggs survival identification with a detection accuracy of up to 99.50% and 99.62% respectively on our test set. The proposed methods performed well, while we did not achieve the classification of weak embryos. Thus, it is going to be attempted to classify fertile eggs, dead eggs, and weak embryos in more detail in future work.

**Author Contributions:** L.G. and Y.H. wrote the paper; Z.X. and J.X. gave guidance in the experiments and data analysis.

**Funding:** This work was supported by the National Natural Science Foundation of China under grant No. 61771340, Tianjin Science and Technology Major Projects and Engineering under grant No.17ZXHLSY00040, No.17ZXSCSY00060 and No.17ZXSCSY00090, and the Program for Innovative Research Team in University of Tianjin (No.TD13-5034).

**Conflicts of Interest:** The authors declare no conflict of interest.

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
