# Peer review of "Fertility Detection of Hatching Eggs Based on a Convolutional Neural Network"

_applsci, doi:10.3390/app9071408_

Round 1

Reviewer 1 Report

Authors presented a method to detect the fertility of hatching eggs by applying the CNN frame work and heartbeat signal of hatching egg. This topic is very interesting. However, this paper should be reviewed again before published.

A. Major revisions:

Comparison to the previous approaches is required. For example, the previous methods they should consider are listed as follows

1. Geng, L.; Yan, T.; Xiao, Z.; Xi, J.; Li, Y. Hatching eggs classification based on deep learning. Multimed. Tools Appl. 2017, 1, 1-12

2. Liu L, Ngadi MO (2013) Detecting fertility and early embryo development of chicken eggs using near infrared hyper spectral imaging. Food. Bio/Technology 6(9):2503–2513

3. Shan B (2010) Fertility Detection of Middle-stage Hatching Egg in Vaccine Production Using Machine Vision. International Workshop on Education Technology and Computer Science. ETCS pp.95–98

4. Paper “Detection of infertile eggs using visible transmission spectroscopy combined with multivariate analysis” at https://www.sciencedirect.com/science/article/abs/pii/S1881836616301434

Please explain why they have to develop a new method while previous ones gave very accuracy result (> 98%).

B. Minor revisions:

There are a lot of error grammars and typos in this article

1. Title, line 2-3

2. Line 15

3. Abstract: line 15-23. Terms such as PhotoPlethysmoGraphy(PPG),E-CNN,SR-CNN, SE-Res, E-CNN, SR-CNN should be defined or explained first before used.

4. Line 106

5. Line 119-120: what do they mean with “the approach of transmissive method”

6. Line 210

7. Line 215

8. Line 353

Author Response

Dear Reviewer:

Thank you for your comments concerning our manuscript entitled

“Detection on the Fertility of Hatching Eggs Based on Convolutional Neural Network”.

Those comments are all valuable and very helpful for revising and improving our paper,

as well as the important guiding significance to our researches.

We have studied comments carefully and have made correction which we hope to meet

with approval. The responses to the your comments are in file "Response to Reviewer 1

Comments".

Reviewer 2 Report

This paper presents a methodology for detecting the fertility of hatching eggs based on the convolutional neural network. Even though authors show the efficiency of the proposed method using experimental results on their own dataset, there are some technical issues as follows:

1) First of all, convolution operators have shown the good performance of 2-dimensional input, i.e., image, however, the input of this paper is 1-dimensional sequence. Why do authors use the convolutional neural network for this task ? (what is the motivation of the proposed method using the convolution operators ?) Furthermore, the shape of filters for convolutional layers should be provided in Table of revision.

2) The novelty of the proposed method is not clear. They propose E-CNN and SR-CNN, however, the architecture of E-CNN is just like the baseline of convolution-based neural network and SR-CNN is just combined with SE-block and residual learning. What is the main difference with previous methods ? Why do authors propose to combine SE-block with residual learning (why is it useful for fertility detection ?) ? Author should clarify this point.

3) To confirm the efficiency of the proposed method, authors need to compare their method with other deep learning-based approaches, e.g., [6] and [8]. Without the performance comparison, it makes the proposed method less convincing. Authors should provide the performance comparison with other deep learning-based approaches in revision.

Author Response

Dear Reviewer:

Thank you for your comments concerning our manuscript entitled

“Detection on the Fertility of Hatching Eggs Based on ConvolutionalNeural Network”. Those comments are all valuable and very helpful for revising and improving our paper,

as well as the important guiding significance to our researches. We have studied comments carefully and have made correction which we hope to meet

with approval. The responses to the your comments are in file "Response to Reviewer 2

Comments".

Round 2

Reviewer 1 Report

All questions have been answered. The manuscript looks like better now. One thing I recommend to be changed is the title "Fertility Detection of Hatching Eggs Based on Convolutional Neural Network". It should be " Fertility Detection of Hatching Eggs using Convolutional Neural Network" or " Convolutional Neural Network Based Fertility Detection of Hatching Eggs" or " Deep Learning Based Fertility Detection of Hatching Eggs".

Reviewer 2 Report

All my concerns are well addressed in revision. Experiments for the performance comparison are good enough to show the efficiency of the proposed method. It is thought that the manuscript is now ready to be published.